# The Pleiades are a cluster of fungal effectors that inhibit host defenses

Fernando Navarrete[1], Nenad Grujic[1], Alexandra Stirnberg[1], Indira Saado[2], David Aleksza[1¤a], Michelle Gallei[1¤b], Hazem Adi[1], André Alcântara[1], Mamoona Khan[3], Janos Bindics[1¤c], Marco Trujillo[4], Armin Djamei[1,2,3¤d]*

1 Gregor Mendel Institute (GMI), Austrian Academy of Sciences (OEAW), Vienna BioCenter (VBC), Vienna, Austria, 2 Leibniz Institute of Plant Genetics and Crop Plant Research (IPK), OT Gatersleben, Stadt Seeland, Germany, 3 Department of Plant Pathology, Institute of Crop Science and Resource Conservation (INRES), University of Bonn, Bonn, Germany, 4 Albert-Ludwigs-University Freiburg, Faculty of Biology, Institute of Biology II, Freiburg, Germany

¤a Current address: Universität für Bodenkultur Wien. Department for Agrobiotechnology (IFA-Tulln). Tulln a.d. Donau, Austria
¤b Current address: Institute of Science and Technology Austria, Klosterneuburg
¤c Current address: Institute of Molecular Biotechnology, Vienna, Austria
¤d Current address: Department of Plant Pathology, Institute of Crop Science and Resource Conservation (INRES), University of Bonn, Bonn, Germany
* djamei@uni-bonn.de

**Data Availability Statement:** All relevant data are within the manuscript and its Supporting Information files.

**Funding:** Research leading to these results has received funding from the European Research

## Abstract

Biotrophic plant pathogens secrete effector proteins to manipulate the host physiology. Effectors suppress defenses and induce an environment favorable to disease development. Sequence-based prediction of effector function is impeded by their rapid evolution rate. In the maize pathogen *Ustilago maydis*, effector-coding genes frequently organize in clusters. Here we describe the functional characterization of the *pleiades*, a cluster of ten effector genes, by analyzing the micro- and macroscopic phenotype of the cluster deletion and expressing these proteins *in planta*. Deletion of the *pleiades* leads to strongly impaired virulence and accumulation of reactive oxygen species (ROS) in infected tissue. Eight of the Pleiades suppress the production of ROS upon perception of pathogen associated molecular patterns (PAMPs). Although functionally redundant, the Pleiades target different host components. The paralogs Taygeta1 and Merope1 suppress ROS production in either the cytoplasm or nucleus, respectively. Merope1 targets and promotes the auto-ubiquitination activity of RFI2, a conserved family of E3 ligases that regulates the production of PAMP-triggered ROS burst in plants.

## Author summary

The genomes of filamentous plant pathogens encode for hundreds of protein-coding effector genes. These secreted proteins target different host components to modify its physiology and promote disease. Effector coding genes usually evolve fast and lack largely functional domains, making the prediction of their function a difficult task. In the maize pathogen *Ustilago maydis*, effector coding genes usually cluster together in the genome.

Council under the European Union's Seventh Framework Programme ERC-2013-STG, Grant Agreement: 335691 (recipient AD) the Austrian Science Fund (FWF): [P27429-B22, P27818-B22, I 3033-B22] (recipient AD) the Austrian Academy of Science (OEAW) (recipient AD) the German Research Foundation under Germany's Excellence Strategy - EXC-2070 – 390732324 (PhenoRob) (recipient AD). The funders had no role in study design, data collection and analysis, decision to publish, or preparation of the manuscript.

**Competing interests:** The authors have declared that no competing interests exist.

Within clusters, new genes are thought to emerge by duplication events followed by quick selection. Here we study the functional relevance of one of these clusters. In the *pleiades*, 8 out of 10 genes function to suppress host defenses, irrespective of their sequence relationship, which constitutes the first functional elucidation of an effector cluster in *U. maydis*. Additionally, we detect the hallmarks of a neo-functionalization event for two paralogs within the cluster of which one effector targets the host proteasomal degradation pathway in the nucleus and the other acts in the plant cytoplasm or the plasmamembrane. Our study shows the functional relevance of effector gene organization and highlights the redundancy, yet mechanistic diversity, in effector functions.

## Introduction

In order to interact with their host plants, pathogenic microbes evolved molecules known as effectors. These secreted molecules (proteins, RNA and small metabolites) manipulate host physiology and development to suppress immune responses and create an environment that promotes the pathogen's proliferation. Following secretion, effector can remain in the space between the plant cells, the apoplast (apoplastic effectors) or translocate into the host cytosol (symplastic effectors) [1,2]. As a result of their secreted nature, effectors are exposed to the host immune system and can induce resistance responses following their recognition. Hence, the evolution of effector coding genes is governed by two major processes, evasion of host recognition and functional optimization, both of which can result in gene lost/gain events, point mutations, or alterations in gene expression [1].

A major problem in effector biology is how to assign function to genes encoding putative effectors. Typically, the genomes of filamentous pathogens code for hundreds or even thousands of effector candidates [1,3,4] yet, for any given pathogen, only a handful have been functionally characterized. Predicting effector gene function from their sequence is hampered by the fact that most of them lack conserved protein domains or homologs beyond closely related species [5]. They are also difficult to study by reverse genetic approaches as many exhibit functional redundancy. Large scale single knockout studies have identified only a few genes with a measurable contribution to virulence [6,7]. This redundancy is believed to counteract recognition by the host immune system. If a single effector targets a given host pathway, losing this effector to avoid host recognition will cause a fitness cost to the pathogen population. However, if multiple effectors target the same pathway, the population can quickly adapt by losing the recognized effector without a fitness cost [1]. For example, many pathogens have multiple LysM effectors that bind chitin in order to avoid host recognition or shield their cellwall from degradation by chitinases [8–10].

Effector encoding genes tend to be enriched in discrete regions of the genome, which show higher sequence diversity within the population in comparison to housekeeping genes. These regions are usually found in sub-telomeric parts of the chromosomes, are enriched in transposon sequences, have low gene density, and/or frequently show presence/absence polymorphisms. All of these features are believed to increase the variability of the effector repertoire in a pathogen population [1].

Smut fungi are a class of pathogens that infect grasses and some dicotyledonous plants, producing high amounts of spores in the floral tissues [11,12]. *Ustilago maydis* infects maize and teosintes, inducing the production of galls in all aerial tissues as soon as 7 days post infection. This feature, together with its genetic tractability, has established the *U. maydis*-maize pathosystem as a model to study biotrophic plant pathogen interactions. In *U. maydis* and other

smuts, a large proportion of effector genes are organized in clusters. These clusters consist of groups of 3 to 26 genes in which related genes are arranged in tandem, likely due to gene duplication events followed by strong natural selection [5,11,13]. Expression of clustered genes is co-regulated; they are upregulated upon infection whereas flanking genes are not [11,14]. Clustered effectors are known to contribute to virulence, although with some degree of redundancy. Individual deletion of seven of these clusters lead to reduced virulence or apathogenic phenotypes [5,11]. However, besides establishing a general role in virulence, reverse genetic studies have not been successful in establishing the function of effector gene clusters.

The plant immune system recognizes two major classes of pathogen-derived molecules, pathogen/microbial associated molecular patterns (PAMPs/MAMPs) and effectors. PAMPs are highly conserved, essential structural components of the pathogen and cannot be modified without large fitness costs. These include bacterial flagellin, fungal chitin and β-glucans [15,16]. The recognition of PAMPs is mediated by a large set of plasma membrane receptors called pattern recognition receptors (PRRs). PAMP binding to PRRs leads to a series of defense reactions that include the rapid production of ROS by membrane-anchored NADPH oxidases (RBOH proteins), increase in cytosolic $Ca^{2+}$ levels, MAP kinase activation, and transcriptional reprogramming. Together, these contribute to PAMP-triggered Immunity (PTI). Consequentially, these signaling cascades are often manipulated by bacterial effectors in order to promote virulence [17]. Likewise, the *U. maydis* effector Pep1 inhibits the activity of host apoplastic peroxidases involved in the production of ROS [18,19].

The ubiquitin proteasome system controls multiple layers of the plant immune system, including PAMP recognition and downstream signaling. Particularly, E3 ubiquitin ligases (E3s) have been shown to regulate the activity and turnover of components involved in immune signaling. In *Arabidopsis thaliana*, the three closely related E3s, PUB22, PUB23, and PUB24 negatively regulate immunity by dampening PAMP recognition [20]. In contrast, the "*Arabidopsis TOXICOS EN LEVADURA*" gene family encodes E3s which are induced by PAMP treatment and contribute to pathogen resistance [21,22]. E3s are frequently targeted by effectors. The *P. infestans* effector Avr3a stabilizes CMPG, an E3 whose degradation is necessary for immune reactions [23]. Most strikingly, the *Pseudomonas syringae* effector AvrPtoB is an E3. It mediates the ubiquitination of several PRRs, leading to their degradation and thereby promoting virulence [24].

Here we present the functional characterization of the *U. maydis Pleiades* / cluster 10A, containing 10 putatively secreted proteins. By expressing these proteins in the plant cell, we show that 8 of the Pleiades inhibit PAMP-triggered immunity, irrespective of their sequence relationship, thereby identifying a PTI-suppressive function of the cluster. We show that neofunctionalization followed gene duplication in the case of the two paralogs, Taygeta1 (Tay1) and Merope1 (Mer1). Finally, we provide evidence that Mer1 targets and modifies the activity of RFI2 homologs, a conserved family of RING E3s that are involved in early immune responses and control of flowering time in plants. *A. thaliana* plants expressing Mer1 show decreased immunity as well as early flowering.

## Results

### The *pleiades*, a cluster of effector genes, encodes for of secreted proteins that contribute to virulence

The *U. maydis pleiades* cluster (Cluster 10A) encodes ten proteins which lack any sequence similarity to known protein domains and lack cysteine residues (which are frequent in apoplastic effectors) beyond their predicted secretion signals. The cluster contains three gene families, A (UMAG_03745, UMAG_03746, UMAG_03747, UMAG_03750) B (UMAG_03748,

UMAG_03749) and C (UMAG_03752, UMAG_037453), based on protein sequence similarity (24% or more, S1 Table). Two genes (UMAG_03744 and UMAG_03751) encode for proteins without homology to other proteins in the cluster (Fig 1A). While none of the *pleiades* show paralogs outside of the cluster, orthologs of all three gene families are well conserved across the sequenced smuts and display high synteny between *U. maydis*, *Sporisorium reilianum*, and *S. scitamineum*, Thus, the gene cluster is conserved among these species (S2 Table).

The transcriptional analysis of the *pleiades* during the various life-stages of *U. maydis* reveals that they are upregulated during the biotrophic phase [14]. Analysis of the Pleiades with SignalP 5.0 [25] predicted the presence of secretion signals for all proteins with high confidence, whereas the two immediately neighboring genes to the cluster (UMAG_03743 and UMAG_03754) code for proteins without secretion signals (S3 Table). We verified the secretion of two Pleiades by monitoring the localization of mCherry fusions of Tay1 (UMAG_03752) and Mer1 (UMAG_03753) expressed by *U. maydis* during biotrophic growth in maize. Three to four days post infection (dpi), *U. maydis* expressing SP$_{mer1}$-mCherry-Mer1, SP$_{mer1}$-Mer1-mCherry-3xHA or SP$_{tay1}$-mCherry-Tay1 showed localization of the mCherry signal in the edges and tips of the hyphae (Figs 1B and S1A and S1C). On the other hand, *U. maydis* expressing mCherry-Mer1 (without its predicted secretion signal) showed a diffuse localization of the mCherry signal throughout the whole hyphae (Fig 1B). Plasmolysis of infected maize leaves with mannitol, which expands the apoplastic space, showed that both mCherry-effector fusions were freely diffusible in the apoplast (Figs 1B and S1A and S1C). Western blot from infected maize tissue showed that these proteins were expressed as full lengths (S1B Fig). We next used the *U. maydis* AB33 strain to express Mer1 and Tay1 in axenic culture. AB33 filaments *in vitro* in response to nitrate, mimicking to some degree developmental changes induced during host colonization [26]. Using the strong, constitutive *otef* promoter we found that full length Mer1-3xHA and Tay1-3xHA accumulated in both, cell pellet and culture supernatant fractions, whereas the non-secreted protein Actin was only detectable in the cell pellet fraction (S1D Fig). When using the *tay1* promoter (which is ten times stronger than the *mer1* promoter [14], we could not detect the expression of these proteins *in vitro* (S1D Fig). Taken together, these data show that Mer1 and Tay1 are soluble proteins secreted by *U. maydis* into the biotrophic interphase upon host colonization.

Deletion of the whole *pleiades* cluster has been shown to impair virulence of *U. maydis* [11]. Therefore, we generated deletion strains in the solopathogenic SG200 *U. maydis* background to dissect the specific virulence contribution of individual gene families within the *pleiades*. We generated deletions of the entire gene cluster, individual gene families or all the genes in the cluster except *atl1* (formerly *ten1*, UMAG_03744), since the latter was previously reported to contribute to virulence [27]. Deletion of the whole cluster had the strongest effect on virulence. Plants infected with this strain showed mild disease symptoms like chlorosis and small galls (Fig 1C). Simultaneous deletion of the gene families A, B, C together with *plo1* also showed a considerable reduction in virulence, although not as strong as in the whole cluster deletion. Finally, deletion of family C showed a mild defect in virulence, which could be complemented ectopically by either *tay1* or *mer1*. Deletion of family A or B alone did not show any measurable effect on virulence (Fig 1C). Altogether, our experiments indicate that the *pleiades* contribute to virulence additively and that *atl1* has the strongest impact, although there seems to be some level of redundancy since deletion of family A and B alone did not impact virulence.

## The Pleiades suppress early defense responses

To investigate how the *pleiades* contribute to virulence, we analyzed early host defense responses upon infection with *U. maydis* strain SG200 or its derivative SG200*Δple*. One of the

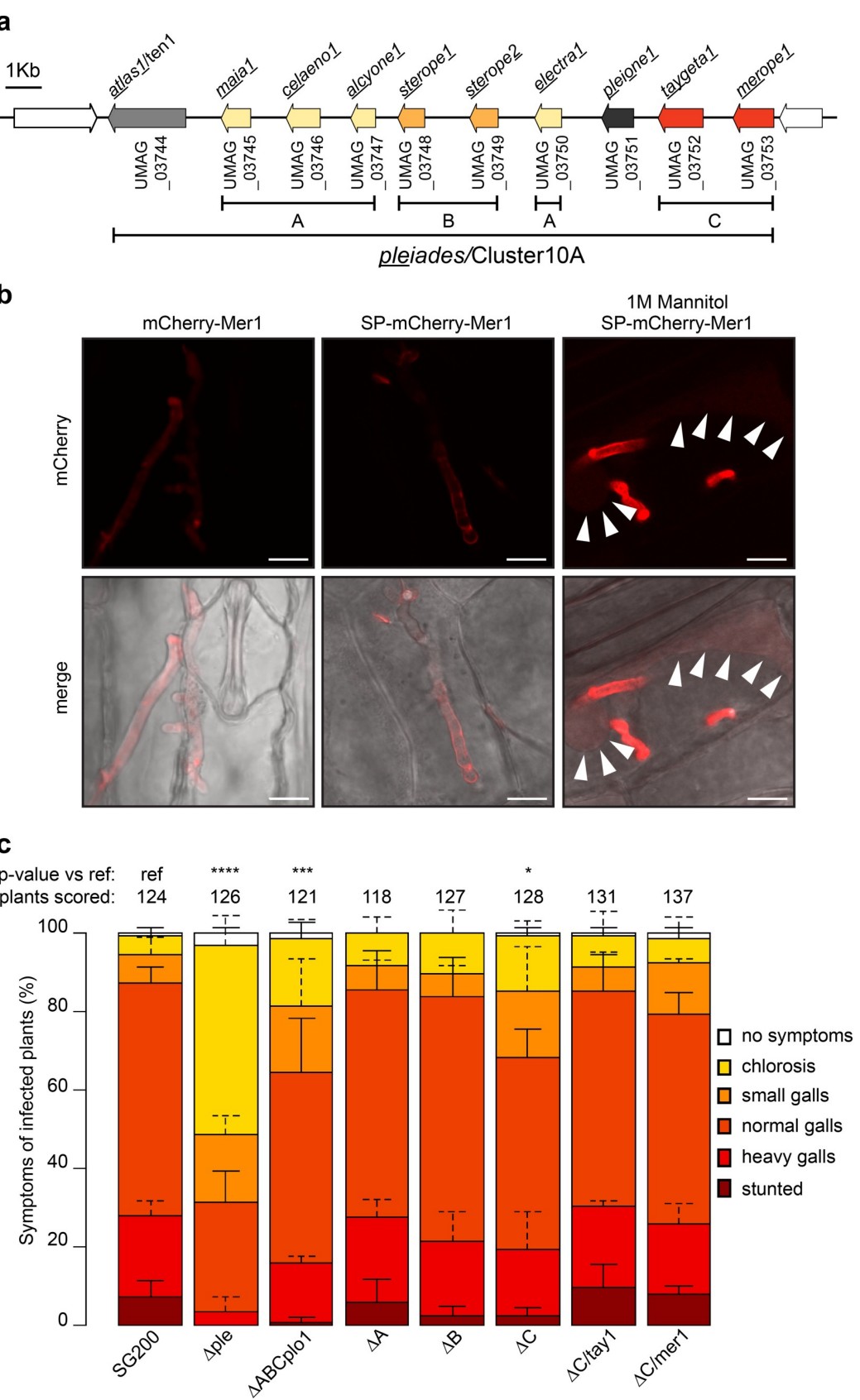

**Fig 1. Pleiades, a cluster of secreted proteins contributes to the virulence of *U. maydis*. a.** Schematic representation of the cluster 10A/Pleiades on chromosome 10 of *U. maydis*. Paralogous genes are represented with the same color and assigned to different families (A, B, C). Genes neighboring the cluster, which do not contain a predicted secretion signal, are shown in white. **b.** Secretion of Mer1 during maize infection. Left: plants infected with a strain expressing mCherry-Mer1 (without its secretion signal) show a diffuse mCherry signal, without accumulation in the periphery of the hyphae. Center: plants infected with a *U. maydis* strain expressing SP$_{Mer1}$-mCherry-Mer1 show mCherry signal mainly at hyphal tip and periphery. Right: plants infected with a strain expressing SP$_{Mer1}$mCherry-Mer1 and plasmolyzed with mannitol show accumulation of the mCherry signal in the hyphae as well as the apoplastic space. Arrowheads: plasma membranes of plasmolyzed maize cells. Upper panels: mCherry fluorescence, lower panels: bright field-mCherry merge. Scale bar = 10 µm. Pictures were taken 3–4 dpi. **c.** Disease symptom scoring of maize seedlings infected with *U. maydis*. SG200 (progenitor strain) or its derivative strains harboring complete or partial deletion of the *pleiades* cluster and their ectopic complementations were infected in seven-day old seedlings, and disease symptoms were rated 12 dpi. Data represent mean ± SD from three independent experiments, n = total number of scored plants. Significant differences between strains were analyzed by the Fisher's exact test with Benjamini-Hochberg correction for multiple comparisons (*p<0.05, ** p<0.01, *** p<0.001, **** p<0.0001).

first signaling and defense responses that plants activate upon recognition of invading microbes is the accumulation of ROS in the apoplastic space, which is usually suppressed by effectors from virulent pathogens [15,16]. We assessed the production of ROS at the infection sites by staining plants 36 hours post infection with diamino-benzidine (DAB), which forms a brown precipitate in the presence of $H_2O_2$ [28,29] and examined them by widefield and fluorescence microscopy. Leaves infected with SG200Δ*ple* showed strong DAB precipitation that accumulated around the invading hyphae, whereas SG200-infected leaves were hardly stained. Hyphae appeared mostly transparent by widefield microscopy (Fig 2A). Hyphae that did not show DAB precipitation could be stained by the chitin-binding WGA-Alexa Fluor 488 dye (WGA-AF488), indicating that all examined areas were colonized by *U. maydis* (Fig 2A).

To complement $H_2O_2$ visualization in the *U. maydis*-maize pathosystem and to test whether the Pleiades have a direct impact on the early PTI response, we expressed each of the proteins (without their predicted secretion signal) in *N. benthamiana* and tested their ability to suppress PAMP-triggered ROS production. Plants expressing each of the *pleiades* were treated with the PAMPs flg22 or chitin and ROS production was monitored overtime using a luminol-based assay [30]. The ROS-burst response in plants expressing mCherry cloned in the same vector used for effector expression was used as reference control. Strikingly, all Pleiades except Plo1 were able to inhibit the PAMP-triggered ROS burst compared to the control (Figs 2B and S2). Expression of Ste2 could not be detected in *N. benthamiana* (S2E Fig). Most proteins inhibited the oxidative burst irrespective of the PAMP used, except Atl1 which was specific for flg22 and Cel1 which was specific for chitin. Taken together, our results suggest that the Pleiades are secreted into the biotrophic interphase and contribute to inhibition of PAMP-triggered ROS burst during host colonization. The fact that the Pleiades inhibit PAMP-triggered ROS burst when expressed in the cytosol of *N. benthamiana* suggests that they might be translocated effectors and their molecular targets might be conserved across monocots and dicots.

## The paralogs Tay1 and Mer1 target different cellular compartments

Since Tay1 and Mer1 had the strongest and most consistent effect on PAMP-triggered ROS production in *N. benthamiana* (Figs 2B and S2C), we focus further experiments on these two proteins.

We employed the recently developed foxtail mosaic virus (FoMV) vectors [31] to express Tay1 and Mer1 in maize. The ROS-burst response in plants expressing GFP cloned in the same vector was used as reference control. Expression of Tay1$_{28-398}$ and Mer1$_{23-341}$ (without their secretion signals) lead to a reduced ROS burst response compared to the GFP control (Fig 3A). This data, together with our previous observation on maize plants infected with

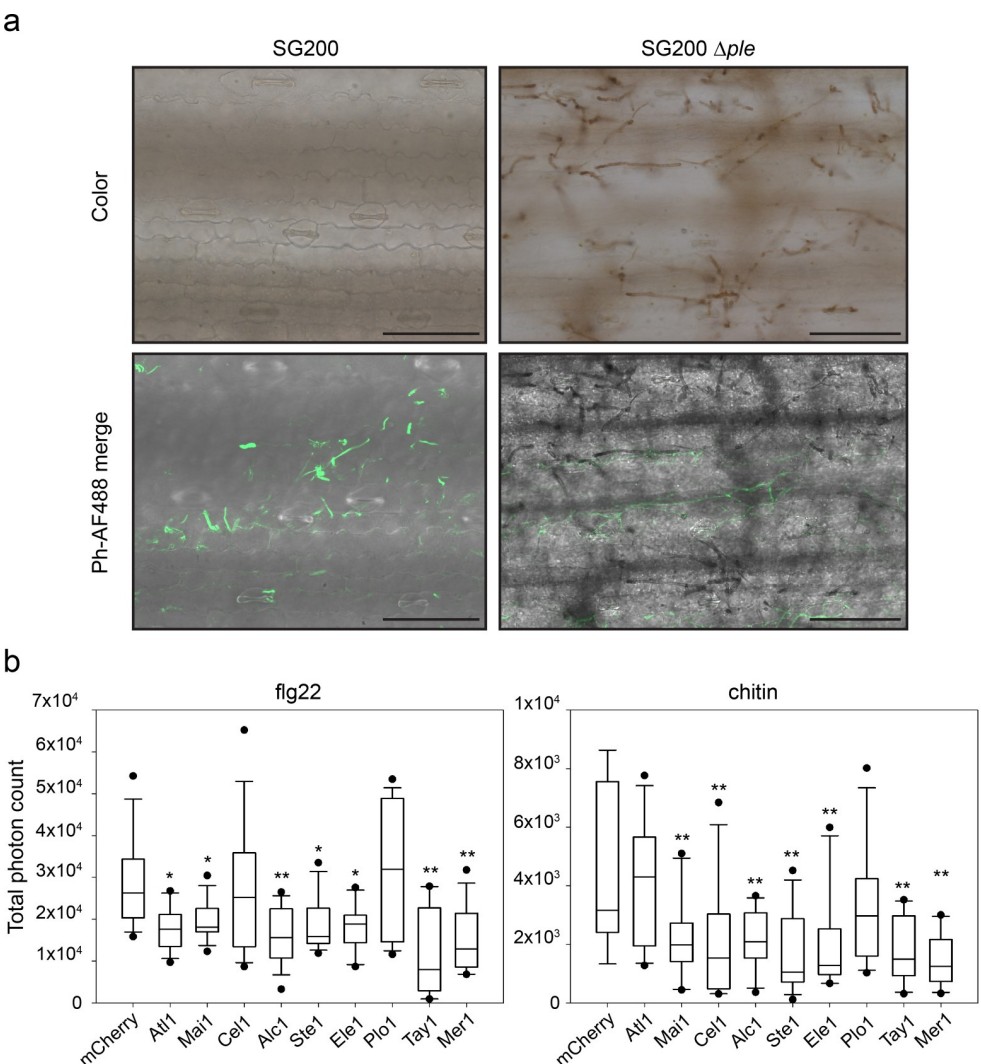

**Fig 2. Pleiades contribute to the suppression of early defense responses. a.** Accumulation of $H_2O_2$ in infected leaves visualized by diamino benzidine (DAB) staining. Left panels: *U. maydis* SG200 colonizes the host tissue without showing strong DAB precipitation. Right panels: *U. maydis* SG200 *Δple* shows pronounced precipitation of DAB around the hyphae upon colonization of the host. Upper panels: color pictures showing the brown DAB precipitate. Lower panels: Phase contrast/Alexa Fluor 488 merge. WGA-Alexa Fluor 488 (which binds chitin) was used as a counter stain to show the presence of hyphae not stained by DAB since the precipitate covers and mask chitin from being bound by WGA. Samples were collected 36h post infection. Scale bar = 100 μm. Color pictures and the Fluorescent/Phase contrast pictures are taken with different cameras; therefore, the fields of the upper and lower panels don't overlay perfectly. **b.** PAMP-triggered ROS burst in *N. benthamiana*. Transient expression of *pleiades* (without their predicted secretion signal) leads to reduction of the oxidative burst triggered by flg22 (left) or chitin (right). Plants expressing mCherry were used as the reference control. Total photon counts over 40 (flg22) or 30 minutes (chitin) are shown as box plots. Data is a pool of three (flg22) or four (chitin) independent experiments, n = 15 or 12 plants respectively. Significant differences between proteins were analyzed by ANOVA with Benjamini-Hochberg correction for multiple comparisons (* $p<0.05$, ** $p<0.01$).

*U. maydis Δple* showing strong accumulation of $H_2O_2$ around invading hyphae, suggest that the Pleiades Tay1 and Mer1 function to suppress PAMP-triggered ROS burst in the host plant.

To further characterize ROS-burst suppression by Tay1 and Mer1, we analyzed their sub-cellular localization. We constructed mCherry fusions of Tay1 and Mer1 and expressed them in maize epidermal cells by biolistic bombardment as well as in *N. benthamiana*, by *Agrobacterium*-mediated transformation. In both cases, GFP-nuclear localization signal (NLS) was

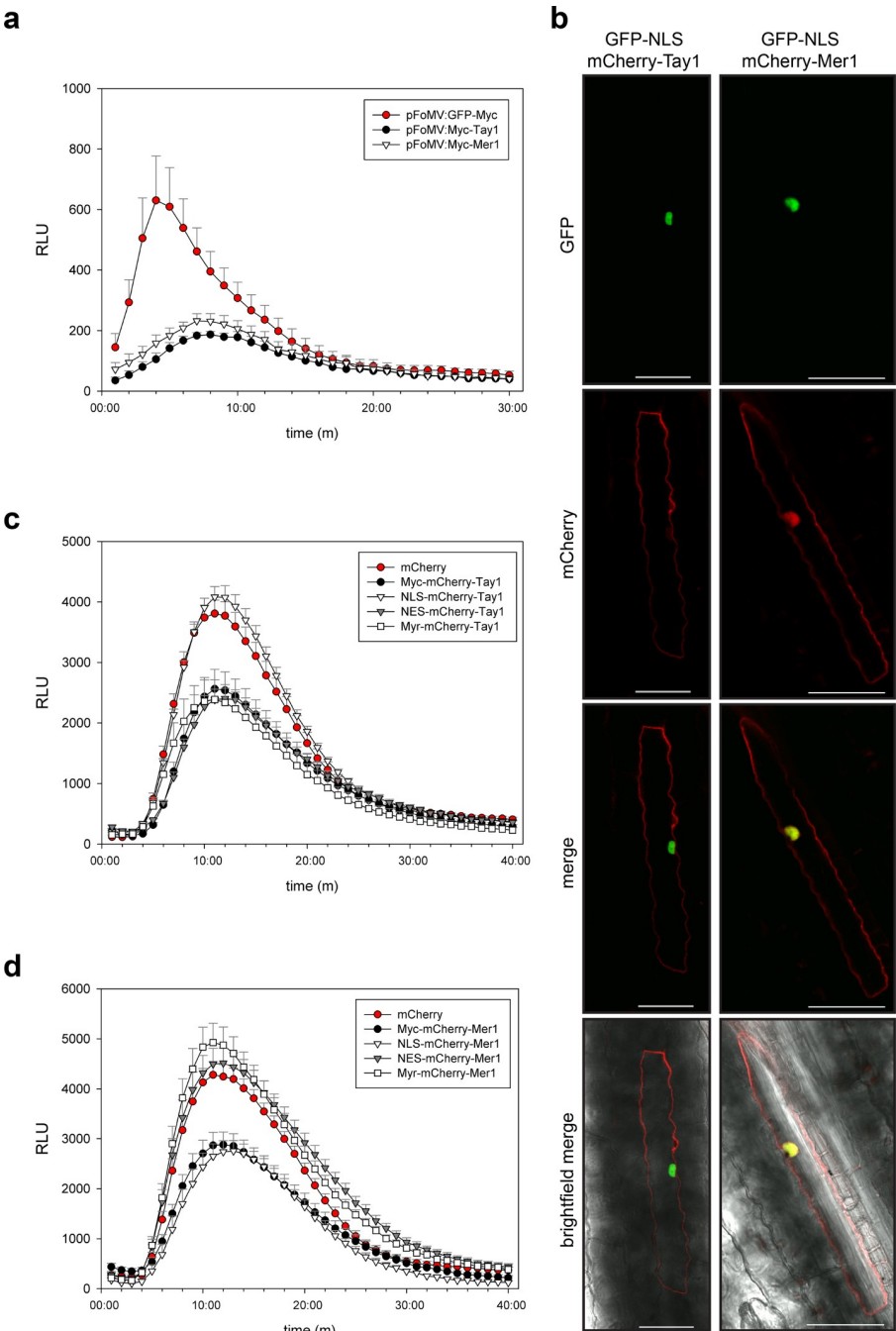

**Fig 3. Tay1 and Mer1 are functional in maize and target different cell compartments. a.** PAMP-triggered ROS burst in maize. Transient expression of Tay1$_{28-398}$ and Mer1$_{23-341}$ in maize with the viral vector pFoMV leads to reduction of the oxidative burst triggered by flg22. **b.** Expression of mCherry- Tay1$_{28-398}$ and mCherry- Mer1$_{23-341}$ in maize epidermal cells by biolistic bombardment. mCherry-Tay1$_{28-398}$ (left panels) localizes preferentially to the cytoplasm whereas mCherry-Mer1$_{23-341}$ (right panels) localizes to the cytoplasm and nucleus. GFP-NLS was co-expressed as a nuclear marker. Scale bar = 50 μm. **c. d.** Sub-cellular localization affects the ROS burst inhibiting activity of Tay1 and Mer1 differentially. Plants expressing mCherry were used as reference control showing maximum ROS-burst and plants expressing Myc-mCherry-Tay1$_{28-398}$ (c) or Myc-mCherry-Mer1$_{23-341}$ (d) were used to monitor their intrinsic ROS-burst inhibitory activity without miss localization. Curves show flg22-triggered ROS burst over 40 minutes. **c.** Removing Tay1 from the nucleus (NES-mCherry-Tay1$_{28-398}$ or Myr-mCherry-Tay1$_{28-398}$) does not affect its inhibitory activity, whereas targeting Tay1 to the nucleus (NLS-mCherry-Tay1$_{28-398}$) abolishes its inhibitory activity. **d.** Removing Mer1 from the nucleus (NES-mCherry-Mer1$_{23-341}$ or Myr-mCherry-Mer1$_{23-341}$) abolishes its inhibitory activity, whereas targeting Mer1 to the nucleus (NLS- mCherry-Mer1$_{23-341}$) does not modify its inhibitory

activity. Data, mean ± SEM, is a pool of three independent experiments, n = 15. Only positive error bars are shown for clarity.

co-transformed and used as a nuclear marker. Confocal microscopy showed that mCherry-Tay1$_{28-398}$ localized primarily to the cytoplasm and almost no mCherry signal was detected in the nucleus (Figs 3B and S3A). On the other hand, mCherry-Mer1$_{23-341}$ was detected in the nucleus as well as in the cytoplasm (Figs 3B and S3B). These results lead us to hypothesize that these paralogs may target different cellular compartments to suppress ROS production. We tested this hypothesis using a mis-localization approach. We fused Tay1 and Mer1 to either NLS-mCherry, NES-mCherry (nuclear export signal) or Myr-mCherry (myristoylation signal, targeting proteins to the inner side of the plasma membrane) and expressed these fusion proteins in *N. benthamiana*. Myc-mCherry fusions of Tay1$_{28-398}$ and Mer1$_{23-341}$ were used as controls since this tag is not expected to affect cellular localization. The localization of the respective mis-localization constructs was visualized by confocal microscopy (S3A and S3B Fig). Expression level and integrity of the fusion proteins were assessed by westernblot using α-mCherry antibodies (S3C Fig). Plants were treated with flg22 and ROS production was monitored over time as described before. Localizing each paralog to different cellular compartments had opposite effects. Fusing Tay1$_{28-398}$ to NLS-mCherry abolished its inhibitory activity on PAMP-triggered ROS-burst (Fig 3C) while fusing Tay1$_{28-398}$ NES-mCherry or Myr-mCherry had no effect with respect with the Myc-mCherry-Tay1$_{28-398}$ control. In contrast, Mer1 showed loss of ROS-burst inhibition activity upon forced cytoplasmic localization (NES-mCherry and Myr-mCherry fusions). Fusing Mer1$_{23-341}$to NLS-mCherry did not affect its inhibitory activity (Fig 3D). Hence, by integrating the microscopy (in maize and *N. benthamiana*) with the ROS-inhibition data upon mis-localization, we propose that Tay1 acts in the host cytoplasm whereas Mer1 acts in the host nucleus.

## Mer1 interacts with RFI2, a family of E3-Ligases

To identify host targets of Tay1 and Mer1 we performed yeast two-hybrid (Y2H) screens against a cDNA library from *U. maydis*-infected maize tissues. Whereas the screen with a binding domain fusion to Tay1 (BD-Tay1$_{28-398}$) did not lead to the identification of any reproducible interactors, screening with BD-Mer1$_{23-341}$ led to the identification of 72 clones growing on high stringency media [-leucine (-L), -tryptophan (-W),—histidine (-H),—alanine (-A)]. Of these, 31 corresponded to proteins with homology to the *A. thaliana* RING domain, E3 red and far red insensitive 2, *RFI2* [32]. To independently verify this interaction, we cloned four of the five predicted RFI2 homologs from maize; the two predicted homologs from *A. thaliana* (we renamed the only characterized homolog At*RFI2A* and its uncharacterized paralog *RFI2*B) and the only predicted homolog from *N. benthamiana* into a prey vector and tested their interaction against BD-Mer1$_{23-341}$. All homologs showed interaction with BD-Mer1$_{23-341}$on intermediate stringency media (-L, -W, -H). Additionally, the homologs ZmRFI2A, ZmRFI2B, ZmRFI2T, and NbRFI2 showed interaction on high stringency media (Fig 4A). ZmRFI2T is a truncated variant, isolated from the cDNA library, which lacks the RING domain (S4A Fig). We also tested whether Tay1 could interact with RFI2 homologs by directed Y2H assays. BD-Tay1$_{28-398}$ was also able to interact with some of the RFI2 homologs tested, although to a much weaker extent compared to BD-Mer1$_{23-341}$(Fig 4A).

To confirm the interaction between family C effectors and RFI2 homologs we performed co-immunoprecipitation assays (Co-IPs). Since RING E3s are known to auto-ubiquitinate, thus mediating their own degradation *in vivo* [33,34], we constructed stabilized versions containing alanine substitutions in the Zn-coordinating residues of the RING domain

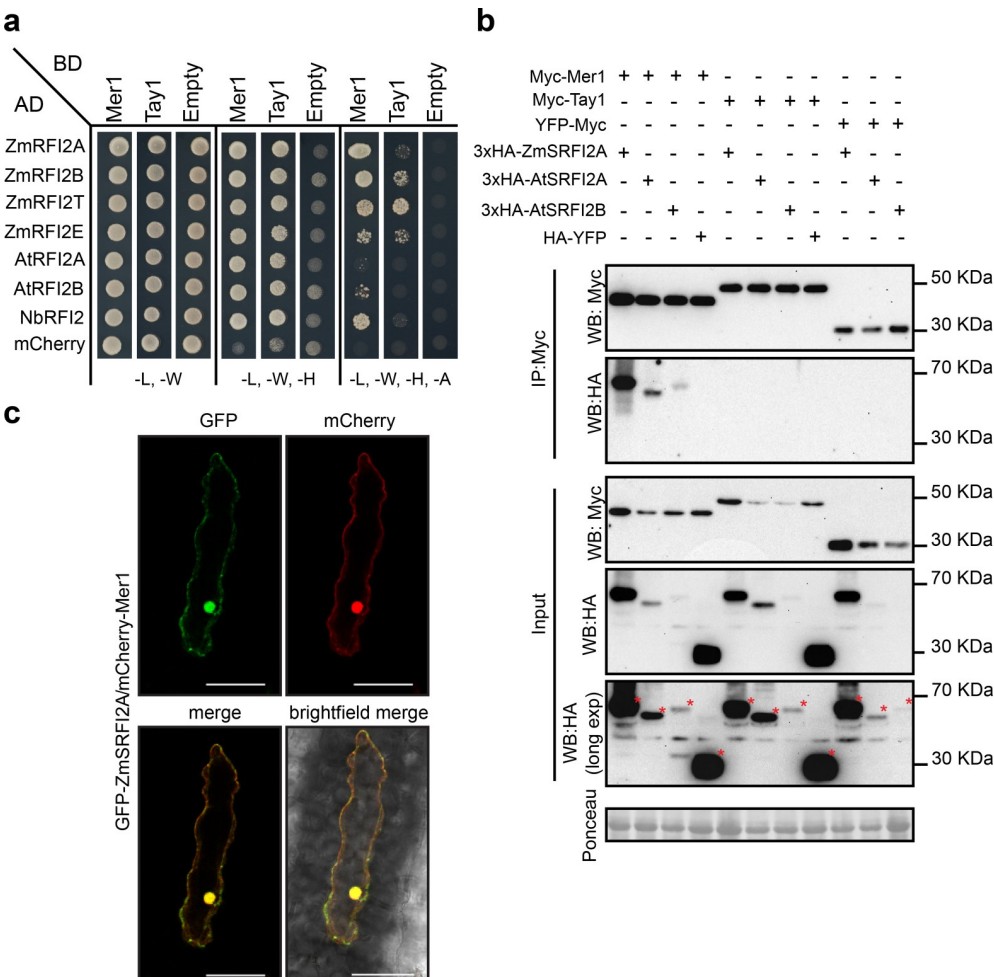

**Fig 4. Mer1 interacts with members of the RFI2 family in the plant nucleus. a.** Y2H assay showing interaction of Mer1$_{23-341}$ and Tay1$_{28-398}$ with RFI2 homologs from maize, *A. thaliana* and *N. benthamiana*. Yeast strains were grown on SD medium lacking indicated amino acids/nucleotides. Growth in media lacking leu (L) and trp (W) is used as transformation control. Growth in media lacking leu (L), trp (W) and his (H) (intermediate selection) or leu (L), trp (W), his (H) and ade (A) (high stringency selection) indicates protein interaction. **b.** RFI2 proteins co-immunoprecipitate with Mer1$_{23-341}$ but not Tay1$_{28-398}$*in planta*. Proteins were co-expressed in *N. benthamiana* by *Agrobacterium* mediated transformation, total proteins were immunoprecipitated with α Myc magnetic beads (IP: Myc) and blotted with specific antibodies. 3xHA-ZmSRFI2A, 3xHA-AtSRFI2A and 3xHA-AtSRFI2B co-immunoprecipitate with Myc- Mer1$_{23-341}$ specifically. Protein loading is indicated by Ponceau staining in the input fraction. (*) indicates the full-length proteins. **c.** mCherry-Mer1$_{23-341}$ and GFP-ZmSRFI2A co-localize in the maize nucleus. Proteins were transiently co-expressed in maize epidermal cells by biolistic bombardment. Scale bar = 50 μm.

(S4A and S4C Fig) which we named "stabilized RFI2" (SRFI2). 3xHA-ZmSRFI2A, 3xHA-AtSRFI2A, or 3xHA-SAtRFI2B were co-expressed with Myc-Mer1$_{23-341}$ or Myc-Tay1$_{28-398}$ in *N. benthamiana*. Proteins were extracted and incubated with α-Myc magnetic beads. All three RFI2 homologs tested were able to co-immunoprecipitate in the presence of Myc-Mer1$_{23-341}$ but not in the presence of Myc-Tay1$_{28-398}$ or YFP-Myc (negative control, Fig 4B). Additionally, HA-YFP was not co-precipitated in the presence of Myc-Mer1$_{23-341}$ or Tay1$_{28-398}$ (Fig 4B). These results show that the interaction between RFI2 homologs and Mer1 is specific.

Since we established earlier that Mer1 but not Tay1 targets the host nucleus (Fig 3), we tested whether RFI2 homologs localize to this cellular compartment. We co-bombarded fluo-rescently labeled proteins into the maize epidermis and verified their localization by confocal

microscopy. GFP-ZmSRFI2A and mCherry-Mer1$_{23-341}$ co-localized in the nucleus and to a lesser extent, in the cytosol (Fig 4C). Co-expression of GFP-ZmSRFI2A, GFP-AtSRFI2A, GFP-AtSRFI2B or GFP-NbSRFI2B with mCherry-Mer1$_{23-341}$ in the epidermis of *N. benthamiana* showed similar results, although the nuclear localization of the E3s was more pronounced in this system (S4D Fig). Consistent with the Co-IP results, co-expression of GFP-ZmSRFI2A, GFP-AtSRFI2A, GFP-AtSRFI2B or GFP-NbSRFI2B with mCherry-Tay1$_{28-398}$ in *N. benthamiana* showed no co-localization (S4E Fig). Taken together, our results show that members of the RFI2 E3 family interact specifically with Mer1 and that this interaction likely happens in the nucleus, since Mer1 only inhibits the PAMP-triggered ROS burst in this compartment. The interaction with maize, *N. benthamiana*, and *A. thaliana* homologs strongly supports for Mer1 our earlier assumption, that the Pleiades' targets are conserved across monocots and dicots.

## Mer1 modifies the ubiquitination activity of RFI2 homologs

In *A. thaliana*, *RFI2*A has been linked to seedling de-etiolation responses and photoperiodic flowering. *rfi2*A plants show early flowering [32,35]. Additionally, the rice homolog *APIP6*, has been linked to immunity [36]. To clarify the role of *rfi2* homologs in immunity we took advantage of the genetic tractability of *A. thaliana*. We generated plants expressing Mer1$_{23-341}$ and compared them to the *rfi2A*, *rfi2B*, and *rfi2A/rfi2B* knockouts. *A. thaliana* plants expressing Myc-Mer1$_{23-341}$ showed a reduced PAMP-triggered ROS burst compared to wild-type (Col-0) plants (Fig 5A). This phenotype occurred even at low expression levels of Mer1, indicating that the mechanism of ROS-burst suppression requires low effector amounts (S5C Fig). The single knockouts (*rfi2A* and *rfi2B)* and the double knock out (*rfi2A/rfi2B*) plants also showed a reduced ROS burst, similar to that of Myc-Mer1$_{23-341}$plants, however the double knock out did not show any further ROS burst reduction compared to the single knock outs (Fig 5B). This data, together with evidence from the literature, indicate that RFI2 proteins are necessary for PAMP-triggered ROS production and are likely targets of Mer1.

Since expression of Mer1$_{23-341}$ *in planta* mimicked the ROS burst phenotype of the *rfi2* knockouts, we hypothesized that Mer1 has an inhibitory effect on the E3s. To test this hypothesis, we produced and purified Myc-Mer1$_{23-341}$, MBP-HA-AtRFI2A and MBP-Strep-AtRFI2B from *E. coli* and assayed the effect of Mer1 on the ubiquitination activity of the E3s *in vitro*. Both E3s showed a moderate auto-ubiquitination activity, that was strongly enhanced by addition of Mer1 (Fig 5C). In the case of AtRFI2A, we could detect the higher molecular weight auto-ubiquitination products of the E3 by western blot directly with α-HA and α-Ubiquitin antibodies. In the case of AtRFI2B, we could only detect the ubiquitination products with the α-Ubiquitin antibody. As negative controls, we performed reactions lacking either E1 or E3 enzymes, which showed no ubiquitination products.

As Mer1 promotes auto-ubiquitination of RFI2 homologs *in vitro*, we tested the effect of Mer1 on the stability of recombinant AtRFI2A by a cell free degradation assay using native protein extracts from *A. thaliana*. Incubation of recombinant MBP-HA-RFI2A with leaf extracts from Col-0 did not lead to significant degradation over a period of 2 h. In contrast, incubation of MBP-HA-RFI2A with extracts of Myc-Mer1$_{23-341}$ expressing plants lead to partial degradation of the protein, which was detected 30 min after the start of the reaction, and this effect was abolished in the presence of the proteasomal inhibitor MG132 (S6 Fig).

*A. thaliana* plants expressing Mer1$_{23-341}$ also showed an early flowering phenotype compared to their Col-0 background when grown under long day conditions (S5A and S5B Fig). In contrast, *rfi2A* and *rfi2B* plants did not flower early, while the *rfi2A/rfi2B* double knockout flowered slightly late (S5 Fig).

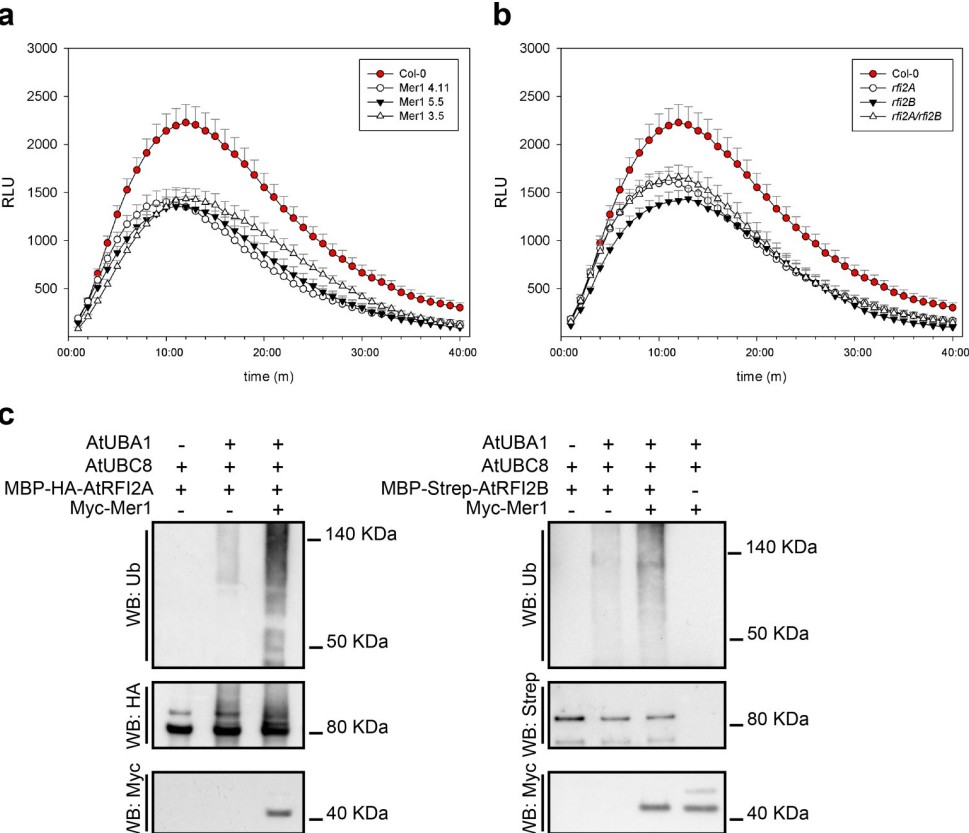

**Fig 5. Mer1 affects the activity of AtRFI2 homologs. a.** PAMP-triggered ROS burst in *A. thaliana*. 35S:Myc-Mer1$_{23-341}$ lines show reduced oxidative burst upon flg22-treatment compared to their Col-0 background. **b.** Both rfi2 homologs of *A. thaliana* contribute to the flg22-triggered ROS burst response. Either single mutant or the double mutant show reduced oxidative burst compared to their Col-0 background. Only positive error bars are shown for clarity. All ROS burst measurements were done simultaneously, panels a and b are shown separately only for clarity. Data, mean ± SEM, is a pool of four independent experiment, n = 20. **c.** Mer1 promotes the auto-ubiquitination activity AtRFI2 homologs *in vitro*. Proteins were affinity-purified from *E. coli*, used to perform ubiquitination reactions and the results were assayed by blotting with specific antibodies. Addition of Mer1$_{23-341}$ promotes the ubiquitination of AtRFI2A (left) or AtRFI2B (right). Reactions excluding E1 or E3s were used as negative controls.

Taken together, our results indicate that the effector Mer1 targets the RFI2 family of E3 ubiquitin ligases which contribute to PAMP-triggered ROS burst. Mer1 inhibits their activity by promoting auto-ubiquitination, leading to higher proteasomal degradation of the E3s and decreased PAMP-triggered ROS-burst. However, we do not discard the possibility that Mer1 has additional host targets.

## Discussion

The analysis of the *U. maydis* genome revealed that a significant number of putative effector genes, whose expression is induced upon host infection, are physically clustered in the genome [11]. Similar to prokaryotic operons, gene clusters are commonly found in diverse fungi to co-regulate functionally connected genes [37–39]. As no functional characterization of an *U. maydis* effector gene cluster has been reported, a common role for these clusters during biotrophy was based on assumptions [11]. Here, by analyzing the phenotype of the *pleiades* mutant at a microscopic level and expressing its encoded proteins *in planta*, we show that eight clustered effectors share the ability to suppress PAMP-triggered immunity independent of their

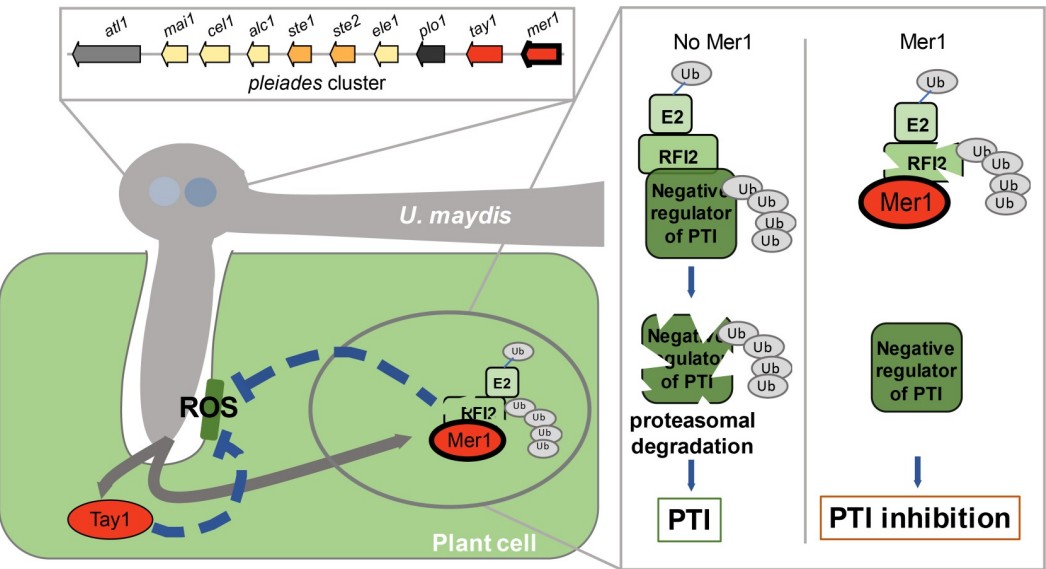

**Fig 6. Working model on the mode of action of the Pleiades.** Left side: Infection of maize by *U. maydis* triggers RBOH-mediated production of Reactive Oxygen Species (ROS)in plant cells around invading hyphae. To counteract this early defense response, *U. maydis* secretes the Pleiades, a heterogeneous group of effectors whose genes are clustered in the genome. Paralogous genes are represented with the same color. Most Pleiades inhibit PAMP-triggered ROS-burst. Tay1 (red oval), inhibits the production of ROS from the cytoplasm. Mer1 (simple red oval with black border), targets the host nucleus, where it promotes auto-ubiquitination and degradation of RFI2s, a family of E3s that regulate the production of. Right side: Hypothetical mode of action of RFI2s. In the absence of Mer1, RFI2s promote the degradation of negative regulators of ROS production. Mer1 leads to the accumulation of the negative regulators and inhibition of ROS production.

sequence relationship. This finding raises the question of whether other clustered effectors also share redundant functions. Moreover, we have increased our understanding of effector functions in smut fungi, where only six effectors have been functionally characterized [19,40–44], and only three are translocated and function within the host cytosol [40,42,43]. Our results and a working model on how the *pleiades* affect the interaction between *U. maydis* and maize are summarized in Fig 6.

Although we have not demonstrated translocation of the Pleiades into host cells directly, our data supports that the Pleiades are symplastic effectors. First, none of the mature Pleiades have cysteine residues, considered a hallmark of apoplastic effectors. Disulfide bond formation occurs under the oxidative conditions of the apoplast, stabilizing protein structures [45,46]. Additionally, the fact that infected plants show higher $H_2O_2$ levels around hyphae of the *Δple* mutant than around those of the progenitor strain agrees with the suppression of PAMP-triggered ROS-burst upon expression of the Pleiades in the plant cytosol. Moreover, we have found that Mer1 promotes the auto-ubiquitination and de-stabilization of RFI2 homologs, a conserved family of E3s that regulate ROS production. Since we have shown these E3 ligases to be positive regulators of PTI, they likely act by promoting the degradation of negative regulators of this process. The identification of RFI2 homologs as effector targets is consistent with the concept that effectors tend to target hubs of the immune system [47,48]. The effector Avr-Piz-t from *Magnaporthe oryzae* was shown to be translocated into rice cells, where it targets a member of the RFI2 family, APIP6 [36]. The fact that two effectors from unrelated pathogens target the same E3 homologs highlights its importance as an essential component for host defense. In an independent study, it was shown by immunogold-labelling and electron microscopy that one of the Pleiades, Atl1/Ten1, is translocated into maize cells during infection by *U. maydis* [27]. The same study further suggests that Atl1 interacts with the PP2C phosphatase

ZmPP26. PP2C phosphatases have been shown to regulate immunity [17,49,50]. Thus, based on our findings and previous studies, we postulate that the Pleiades are likely translocated effectors that act in the host cytosol. The mechanisms that lead to translocation of fungal effectors into host cells are still under debate, and for *U. maydis* remain completely unknown [51–53]. The identification of a critical mass of experimentally validated translocated effectors will help to elucidate commonalities that could be used for predicting their translocation into host cells based on sequence features.

Using heterologous plant systems to study effector function introduces the problem that host-specific factors might be missed. On the other hand, effectors that target highly conserved pathways can be identified. Bearing this in mind, most members of the Pleiades likely target conserved components of the PAMP-triggered immunity in plants, which we have demonstrated for Mer1. On the other hand, Plo1 which did not suppress ROS-burst in *N. benthamiana*, might target a maize specific protein or pathway. Alternatively, it might have not been functional upon expression in in *N. benthamiana* or may simply have a different function. Nevertheless, the Pleiades could be used as tools to identify and dissect conserved elements of the plant immune system.

Our work provides a prime example of functional redundancy by demonstrating that eight clustered effectors collectively inhibit PAMP-triggered ROS-burst, where the pathogen relies on a battery of tools to ensure that critical defense responses are efficiently suppressed during host colonization. This redundancy might allow subpopulations of *U. maydis* to lose some of the *pleiades* in order to avoid recognition by particular host genotypes, while still maintaining the ability to suppress PAMP-triggered ROS burst [1]. Alternatively, there might be some sub-specialization amongst the Pleiades. Indeed, some of the Pleiades show a certain degree of specialization. Cel1 and Atl1 inhibit ROS burst only when triggered by chitin or flg22, respectively. In the case of Atl1, the fact that a fungal effector specifically inhibits flg22 dependent immunity might indicate that it targets BAK1-dependent pathways, which is necessary for the perception of many PAMPs, including flg22 but not chitin [54–56]. Further support for functional specialization of the *pleiades* comes from analysis of their expression pattern, which varies considerably regarding in their timing and tissue specificity [57].

Considering that in smut fungi effectors frequently emerge by gene duplication followed by rapid diversification [5,13], it is likely that a similar evolutionary history favored the appearance of new functional variants within the *pleiades*. For the paralogs Tay1 and Mer1, which are encoded by adjacent genes in the cluster and share 31% identity at the aminoacid level, our mis-localization studies showed that they suppress ROS production in different sub-cellular compartments (cytoplasm or nucleus). Although Tay1 was able to bind to some RFI2 homologs in Y2H assays, we could not verify this interaction by Co-IP nor did Tay1 co-localize with the E3s *in planta*, Therefore, RFI2 homologs seem to be targets of Mer1 specifically. Furthermore, *tay1* is widespread across smuts whereas *mer1* is restricted to the closely related *U. maydis*, *S. reilianum*, and *S. scitamineum*, where *tay1* and *mer1* are neighboring genes (S2 Table). Thus, *tay1* and *mer1* show the characteristics of a neofunctionalization event.

Finally, we showed that Mer1 targets RFI2 homologs to promote early flowering in addition to suppress immunity. Although at*rfi2A* has been shown to negatively regulate flowering time in the *A. thaliana* accession Ws-0 [32] neither At*rfi2A*, At*rfi2B*, nor the At*rfi2A*/At*rfi2B* double mutant showed an early flowering phenotype in the Col-0 accession. However, these two accessions are notoriously different regarding their flowering behavior [58], which might indicate the presence of additional factor(s) that need to be inactivated in Col-0 in order to express the early flowering phenotype. Nonetheless, an effector that dampens immunity while simultaneously promoting flowering would be a great advantage for smuts, which usually sporulate

only in the host floral tissues. This is in line with the concept that effectors frequently target regulatory nodes to shift the balance from immunity to growth and development [2].

In conclusion, we have shown that the Pleiades, a heterogeneous group of proteins, whose genes cluster in the genome of *U. maydis*, share the ability to suppress PAMP triggered immunity. Considering the evidence provided, that the Pleiades act through different mechanisms makes them valuable starting points to identify conserved and possibly novel players of PAMP-triggered immunity in plants.

## Materials and methods

### Gene accession numbers

*atlas*1: UMAG_03744, *maia*1: UMAG_03745, *celaeno*1: UMAG_03746, *alcyone*1: UMAG_03747, *sterope*1: UMAG_03748, *sterope*2: UMAG_03749, *electra*1: UMAG_03750, *pleione*1: UMAG_03751, *taygeta*1: UMAG_03752, *merope*1: UMAG_03753, Zm*rfi2*A: Zm00001d037596, Zm*rfi2*B: Zm00001d009837, Zm*rfi2*T: Zm00001d040789, Zm*rfi2*E: Zm00001d052697, At*rfi2*A: at2g47700, At*rfi2*B: at3g05545, Nb*rfi2*: Niben101Scf00369g11013.1.

### Plasmids, cloning procedures and generation of *U. maydis* strains

All plasmids were generated by standard molecular procedures [59]. *E. coli* Mach1 (Thermo Fisher Scientific, Waltham, MS, USA) was used for all DNA manipulations. Plasmids used for expression of effector proteins in *N. benthamiana* (Fig 2B) were generated by Gateway Cloning [60]. For transient virus-mediated overexpression, PV101 [61] was used to generate pFoMV:p19-P2A-mCherry-P2A-Myc-effector and pFoMV:p19-P2A-mCherry-P2A-gfp-myc using NotI-XbaI cloning sites (p19: silencing suppressor, P2A: viral ribosome skipping motif). All other plasmids were generated by the GreenGate system [62]. We generated two GreenGate destination vectors. pECGG, based on a pET backbone, was used for expression of proteins in *E. coli*. pADGG, based on a pGAD backbone, was used as prey vector for yeast two-hybrid assays.

*U. maydis* knock out strains were generated by homologous recombination with PCR-derived constructs [63]. For complementations and protein expression, strains were generated by insertion of p123 derivatives into the *ip* locus [64]. Transformants were verified by southern blot and/or PCR. All plasmids and strains used in this study can be found in S1 File.

### Maize infection assays

*U. maydis* SG200 and its derivatives were used to infect seven-day-old maize seedlings (Early Golden Bantam, Old Seeds, Madison, WI, USA). Syringe-inoculation and symptom scoring (12 dpi) were performed according to [11]. Maize was grown in a temperature-controlled glasshouse (14h light/10h dark, 28°C/20°C). Filamentous growth of *U. maydis* was tested by spotting in potato dextrose agar containing 1% activated charcoal. Experiments were repeated three times.

### Transient expression in maize by biolistic bombardment

Biolistic bombardment was performed according to [40]. Briefly, 1.6μm gold particles were coated with plasmid DNA encoding the indicated constructs under the CaMV35S promoter. Seven-day old maize leaves cv B73 were bombarded using a PDS-1000/HeTM instrument (BioRad) at 900 p.s.i. in a 27 Hg vacuum. Fluorescence was observed by confocal microscopy 18–24 h after transformation. Experiments were repeated three times.

## Confocal microscopy

Confocal microscopy was performed with a Zeiss LSM 700 confocal microscope. GFP was excited at 488 nm using an argon laser. Fluorescence emission was collected between 500–540 nm. mCherry was excited at 561nm and emission was collected between 578–648 nm. Images were processed with ZEN blue 2.3 lite. For visualization of secreted proteins in infected maize, plants were harvested 3–4 dpi and analyzed as described above. Experiments were repeated three times.

## Protein secretion *in vitro*

Secreted proteins from fungal cultures were detected as in [40]. *U. maydis* was grown in CM medium to an O.D.$_{600nm}$ of 0.6–0.8, centrifuged, resuspended in AM medium and incubated for 6 h to induce filamentation. Cultures were centrifuged; supernatant proteins were precipitated with 10% trichloroacetic acid and 0.02% sodium deoxycholate and resuspended in 100 mM Tris pH 8. Cell pellet proteins were extracted by adding SDS loading buffer, a spatula tip of glass beads and vortexing for 10 m. Extracts from cell pellets and culture supernatants were subjected to immunoblotting using α-HA (Sigma-Aldrich, St. Louis, MO, USA) for detection of effector proteins and α-Actin antibodies (Invitrogen, Waltham, MA, USA) for lysis control. Experiments were repeated three times.

## DAB-WGA staining in infected maize tissue

$H_2O_2$ production in infected maize tissues was detected using diamino-benzidine (DAB) (Sigma-Aldrich, St. Louis, MO, USA). Plants were harvested 36h post infection. The 3$^{rd}$ leaf was cut with a scalpel and the bottom part was dipped in DAB solution (1mg/ml, pH 3.8) in the dark at room temperature for 16h. Leaves where de-stained by several washes with ethanol/chloroform (4:1) until chlorophyll was no longer visible. Samples were washed with PBS and chitin was stained with wheatgerm agglutinin coupled to AlexaFluor488 (WGA-AF488 [Invitrogen] 10μg/ml, Tween20 0.02% in PBS) by applying vacuum three times. Visualization of samples was performed by direct observation with a widefield microscope equipped with Apotome2 (Axio Imager.Z2 sCMOS camera, Axiocamcolour camera and Apotome2). DAB was visualized with a color camera or by Phase contrast. Chitin, was visualized by Apotome2 structured illumination with a 480/40nm excitation filter and 525/50nm emission filters. Images were processed with ZEN blue 2.3 lite. Experiments were repeated at least three times.

## Yeast transformation and two-hybrid assays

All yeast protocols were done according to the Yeast Protocols Handbook (Clontech, Mountainview, CA) with minor modifications. Strain AH109 was transformed with bait vectors (pGBKT7 and derivatives) and strain Y187 was transformed with prey vectors (pAD, pADGG and derivatives) by the LiAc/PEG method. Transformants were verified for the presence of the corresponding plasmid by PCR. For spotting assays, growth of diploids in intermediate or high stringency media 4 days post-inoculation indicated positive interactions. Experiments were repeated two times.

## Plant growth conditions, *A. thaliana* lines and flowering time experiments

*N. benthamiana* and *A. thaliana* were grown in controlled short-day conditions (8h light/16h dark, 21˚C) in "Einheitserde classic" as substrate (https://www.einheitserde.de/produkte/produktlinien/). The lines used here were created by crossing of At*rfi2*B (SALK_089110C) and At*rfi2*A (SAIL_1222_B08) and screening for wildtype, single and double knockouts from the

F2 populations (S1 File). Mer1 plants were created by floral dipping using the following construct: 35S:Myc-Mer1$_{23-341}$ (S1 File). For flowering time experiments plants were stratified 3 days at 4˚C in the dark and grown in long day conditions (16 h light/8 h dark, 21˚C/16˚C). Cool white fluorescent lamps were used as light source (64–99 $\mu$mol m$^{-2}$ sec$^{-1}$). Flowering time was determined as total number of rosette and cauline leaves when the stem reached 1 cm. Experiments were repeated three times.

## ROS burst assays in *Nicotiana benthamiana* and *A. thaliana*

5-6-week-old *A. thaliana*, 4-5-week-old *N. benthamiana* plants were used for the assay. *N. benthamiana* was infiltrated with *A. tumefaciens* and incubated for 48h. Leaf disks (4mm) were floated in water o.n. Water was removed, and elicitors were added. flg22 elicitation solution was: Horseradish peroxidase (HRP 10 $\mu$g/ml, Sigma-Aldrich cat# P6782), L-012 (34ug/ml Fujifilm WAKO cat# 120–04891) and flg22 (100nM) in $H_2O$. Chitin elicitation solution was prepared as follows: 50mg of chitin (Sigma-Aldrich cat# C9752) were ground with mortar and pestle in 5 ml of $H_2O$ for 5 min, transferred to a falcon tube, microwaved for 40 s, sonicated for 5 min, centrifuged at 1800 g for 5 min, supernatant was transferred to a new tube, vortexed for 15 min and stored at 4˚C. Before use, the suspension was diluted 1:1 in $H_2O$ and supplemented with HRP and luminol (34 $\mu$g/ml Sigma-Aldrich cat# 123072). ROS production was monitored by luminescence over 30–40 minutes in a microplate reader (Synergy H1, BioTek). At least three plants per construct/genotype were used in each experiment. Experiments were performed at least three times.

## Transient protein expression and ROS burst assays in maize

Protein expression in maize was achieved using plasmids derived from the FoMV infectious clone described earlier (61). Plasmids were bombarded in 6–7 day-old B73 whole-seedlings and plants were returned to the growth chamber. Nine days post-bombardment, samples were taken and mCherry fluorescence was monitored as an indication of viral spread. Leaf disks from areas with positive mCherry signal were floated o.n. in water and ROS was measured as above. Experiments were repeated twice.

## Protein production in *N. benthamiana* and Co-immunoprecipitation

For *in vivo* Co-IP assays, *A. tumefaciens* GV3101 (pSoup) carrying the expression constructs were grown o.n. in LB supplemented with the appropriate antibiotics at 28˚C, centrifuged and resuspended in ARM buffer (*Agrobacterium* resuspension medium, 10 mM MES-NaOH pH 5.6, 10mM $MgCl_2$, 150 $\mu$M Acetosyringone) to an $OD_{600nm}$ of 0.2 and incubated for 3h at RT. Cultures carrying the appropriate constructs were then mixed 1:1 and infiltrated in *N. benthamiana* with needless syringe. Plants were incubated for 60 h, frozen in liquid $N_2$ and total proteins were extracted from 450 mg of tissue in 2 ml IP buffer: HEPES 50 mM pH7.5, NaCl 100 mM, Glycerol 10%, EDTA 1 mM, Triton X-100 0.1%, PMSF 1 mM and 1 protease inhibitor tablet/50 ml (Roche cOmplete EDTA free cat# 05056489001). Extracts were cleared by centrifugation 10min at 20000g three times. Proteins were immunoprecipitated by adding 30 ul of $\alpha$-c-Myc magnetic beads ($\mu$MACS Anti-c-myc MiltenyiBiotec, cat# 130-091-284) and incubated for 2 h at 4˚C with rotation. Samples were washed 4 times with 300 $\mu$l IP buffer, eluted by adding 50 $\mu$l of 2x SDS loading buffer at 95˚C. 10–15 $\mu$l of extracts were analyzed by by Western blot with $\alpha$-c-Myc (Sigma-Aldrich, St. Louis, MO, USA) or $\alpha$-HA (Sigma-Aldrich, St. Louis, MO, USA) antibodies. Experiments were repeated two times.

## Protein production in *E. coli*, ubiquitination assays and cell free degradation assays

Recombinant protein production was done as follows: His-AtUBA1 and His-AtUBC8 were produced in *E. coli* strain BL21 AI (ThermoFisher). All cultures were grown in LB Ampicillin at 37°C to an $OD_{600nm}$ of 1, cold shocked on ice for 20 m and protein production was induced by adding L-Arabinose (0.2%), or L-Arabinose plus ethanol (1.5%) for UBA1. Cultures were further incubated for 5h at 22°C and pelleted. All remaining proteins (were produced in *E. coli* BL21 pLys. Cultures were grown in LB Spectinomycin at 37°C to an OD600nm of 0.6 and protein production was induced by adding IPTG (0.5 mM). Cultures were further incubated 3h at 37°C or o.n. or at 22°C (His-Myc-Mer1$_{23-341}$) and cell pellets were frozen at -80°C. The version of Mer1 used here was codon optimized for *E. coli*.

Ubiquitination reactions were done in 45 µl as follows: Tris pH 7.5 50 mM, $MgCl_2$ 5mM, KCl 25 mM, $ZnCl_2$ 50 µM, DTT 0.25 mM, ATP 5 mM, Ubiquitin 3 µg (Sigma-Aldrich, cat# U6253), AtUBA1 100 ng, AtUBC8 250 ng, AtRFI2A 250 ng, AtRFI2B 250 ng, Mer1 250 ng. Reactions were incubated at 30°C for 2 h and stopped by adding 15 ul of 4X SDS loading buffer and incubated at 95°C for 5 min. Analysis of the reactions was done by western blotting with StrepTactin-HRP Conjugate (Bio Rad, cat # 1610380), α-c-Myc (Sigma-Aldrich, St. Louis, MO, USA), α-HA (Sigma-Aldrich, St. Louis, MO, USA) and α-Ubiquitin (P4D1, Abcam cat# ab139101) antibodies. Experiments were repeated three times.

Cell-Free degradation assay was performed according to [65]. Briefly, leaves from 5-week-old *A. thaliana* (Col-0 and Mer123-341) were ground to a fine powder in liquid $N_2$. Proteins were extracted in 25 mM Tris-HCl pH 7.5, 10 mM NaCl, 10 mMMgCl$_2$. Extracts were cleared by centrifuging two times for 10 min at 14000g and 4°C. Protein concentration was determined by the Amido-Black assay and all extracts were adjusted to the same concentration with extraction buffer. Reactions were performed in 200µl and contained 50–100 ng of recombinant His-MBP-HA-AtRFI2A, 10 mM ATP and 50 µM MG132 or DMSO. ATP and DMSO were added 30 min prior to the start of the reaction. Reactions were incubated at 30°C for 2 h and samples were taken at the indicated intervals. Recombinant protein abundance was determined by western blot with α-HA antibodies (Sigma-Aldrich, St. Louis, MO, USA).

## Statistical analyses

Maize infection assays were analyzed by the Fisher exact test in R, as described by [66]. All other statistical analyses were performed with GraphPad Prism 8.0. Data from ROS bursts and cell-free degradation assays was analyzed by ANOVA with Benjamini-Hochberg correction for false discovery rate. Flowering time data was analyzed by ANOVA, Tukeys. Statistical significance was evaluated at the level of $p < 0.05$.

## Supporting information

**S1 Fig. Secretion of Tay1 and Mer1.** a. Secretion of Mer1 during maize infection. Left: Plants infected with the *U. maydis* strain SG200ΔC carrying the construct P$_{tay1}$:SP$_{Mer1}$-Mer1-m-Cherry-3xHA in the ip locus show accumulation of the mCherry signal at the hyphal edges and tips. Right: plants infected with the same strain and plasmolyzed with 1 M mannitol show accumulation of the mCherry signal in the hyphae as well as the apoplastic space. Arrowheads: plasma membranes of plasmolyzed maize cells. Upper panels: mCherry fluorescence, lower panels: bright field-mCherry merge. Scale bar = 10 µm. Pictures were taken 4 dpi. b. The expression and integrity of P$_{tay1}$:SP$_{Tay1}$-Tay1-mCherry-3xHA and P$_{tay1}$:SP$_{Mer1}$-Mer1-m-Cherry-3xHA during maize infection was monitored by western blot with α-HA antibodies.

Secreted mCherry-3xHA is included for size reference. Tissue was collected 4dpi. c. Secretion of Tay1 during maize infection. Left: Plants infected with the *U. maydis* strain SG200ΔC carrying the construct $P_{tay1}$:$SP_{Tay1}$-mCherry-Tay1 in the ip locus show accumulation of the mCherry signal at the hyphal edges, tips and cell-to-cell crosses. Right: plants infected with the same strain and plasmolyzed with 1 M mannitol show accumulation of the mCherry signal in the hyphae as well as the apoplastic space. Arrowheads: plasma membranes of plasmolyzed maize cells. Upper panels: mCherry fluorescence, lower panels: bright field-mCherry merge. Scale bar = 10 μm. Pictures were taken 3–4 dpi. d. Secretion of Tay1 and Mer1 in axenic culture. Constructs harboring $SP_{Tay1}$Tay1-3×HA or $SP_{Mer1}$Mer1-3×HA were expressed in the strain AB33 under the tay1 or otef promoter. Total proteins were extracted from the pellet and secreted proteins were precipitated from the culture supernatant. The extracts were subjected to western blot with α HA or α Actin antibodies. Tay1-3×HA and Mer1-3×HA could be detected in the pellet and supernatant fractions only when the expression was driven by the strong synthetic otef promoter. Actin could only be detected in the pellet fraction. e. Filamentation of *U. maydis* strains used in Fig 1C. Strains were spotted on PD-charcoal plates and pictures were taken 24h after. All strains retain the ability to filament, a prerequisite for infection. (TIF)

**S2 Fig. PAMP-triggered ROS burst curves.** PAMP-triggered ROS burst curves corresponding to Fig 2B. Left: flg22, right: chitin. a. Curves corresponding to proteins from family A. b. Curves corresponding to proteins from family B. c. Curves corresponding to proteins from family C. d. Curves corresponding to Atl1 and Ple1. Data is mean ± SEM, n = 15 (flg22) or 12 (chitin). Only the upper error bar is shown for clarity. All the measurements were done simultaneously; curves were split according to protein family only for clarity. e. The expression and integrity of the Pleiades was monitored in *N. benthamiana* by western blot with α-P2A antibodies. mCherry was included as reference. Notice that Ste2 was not detected. (TIF)

**S3 Fig. Localization of Tay1 and Mer1 mCherry fusions in *N. benthamiana*.** GFP-NLS was co-expressed as a nuclear marker. a. Myc-mCherry-Tay1$_{28-398}$, NES-mCherry-Tay1$_{28-398}$ and Myr-mCherry-Tay1$_{28-398}$ localize largely outside of the nucleus, whereas NLS-mCherry-Tay1$_{28-398}$ localizes to the nucleus and cytoplasm. b. Myc-mCherry-Mer1$_{23-341}$ localizes to the nucleus and cytoplasm, NLS-mCherry-Mer1$_{23-341}$ localizes to nucleus, whereas NES-mCherry-Mer1$_{23-341}$ and Myr-mCherry-Mer1$_{23-341}$ localize to the cytoplasm. All scale bars = 50 μm. c. The expression of Tay1 and Mer1 mCherry fusions used in a and b was monitored by western blot with α-mCherry antibodies. α Actin western blot was used as loading control. (TIF)

**S4 Fig. RFI2 is a family of E3 ubiquitin ligases whose localization is enriched in the nucleus.** a. Protein alignment of the N-terminus of RFI2 family members shown in Fig 4A. The residues in red background mark the RING domain. Within the RING domain, Zn-coordinating residues 3, 4 and 5 are marked in black background. b. Maximum likelihood, unrooted phylogenetic tree of the RFI2 homologs shown in a. Branch length represents genetic distance according to Kimura 2-parameter. c. Mutations of the Zn-coordinating residues stabilizes RFI2s. α-HA western blot showing the expression of WT, single or triple alanine substitutions of the residues shaded in black in part a. For each protein, the most stable mutant is marked with "S". YFP-3xHA is shown for comparative reasons. α Actin western blot was used as loading control. c. d. Co-localization of Mer1 and Tay1 with different members of the RFI2 family in the epidermis of *N. benthamiana*. mCherry-Mer1$_{23-341}$ co-localizes with

GFP-ZmSRFI2A, GFP-AtSRFI2A, GFP-AtSRFI2B and GFP-NbSRFI2 at the plant nucleus (d), whereas mCherry-Tay1$_{28\text{-}398}$ does not (e). Scale bar = 50 μm.
(TIF)

**S5 Fig. Flowering Phenotype of *A. thaliana* expressing Mer1.** a. Flowering of Col-0 and two independent Myc-Mer1$_{23\text{-}341}$ lines 29 days after planting. Plants were grown in long day conditions (16h l/8h d). Scale bar = 6 cm. b. Quantification of flowering time, assessed as number of leaves at flowering day, is shown as box plots. 35S:Myc-Mer1$_{23\text{-}341}$ plants show early flowering whereas the *rfi2A*, *rfi2B* and *rfi2A*/*rfi2B* knockouts in the Col-0 background do not. One representative experiment is shown, n = 30 plants. Significant differences between lines were analyzed by ANOVA, Tukeys (* $p<0.05$, ** $p<0.01$). c. Expression of Mer1$_{23\text{-}341}$ in *A. thaliana*. Expression was assessed by western blot with α-Myc antibodies, α-Actin was used as loading control. "*" unspecific band.
(TIF)

**S6 Fig. Mer1 destabilizes AtRFI2A.** a. Cell free degradation assay in *A. thaliana*. MBP-HA-AtRFI2A was purified from *E. coli*, incubated in Col-0 or 35S:Myc-Mer1$_{23\text{-}341}$ crude extracts and its stability was monitored over time by western blot with α-HA antibodies. (*) indicates the full-length protein. Ponceau staining was used as loading control. MG132 was used to assess proteasomal activity on the stability of the recombinant protein. b. Protein quantification over time. Data, mean ± SEM, is a pool of four independent experiments. Significant differences were analyzed by Two way- ANOVA with Benjamini-Hochberg correction for multiple comparisons (* $p<0.05$).
(TIF)

**S1 Table. Homology within each of the Pleiades families in *U. maydis*.**
(DOCX)

**S2 Table. Conservation of the Pleiades proteins across different smut species.**
(DOCX)

**S3 Table. Prediction of signal peptides across the *U. maydis* Pleiades.**
(DOCX)

**S1 File. List of plasmids, strains and plant lines used in this study.**
(XLSX)

## Acknowledgments

We thank the GMI/IMBA/IMP core facilities for excellent technical support, especially, BioOptics and Molecular Biology Services and Mathias Madalinski for peptide synthesis. We thank Rothamsted Research Limited and Dr. Kostya Kanyuka kindly providing the vector PV101 for virus-based protein expression in maize. We thank Dr. J. Matthew Watson for proofreading and valuable feedback on the manuscript. We thank Martin A. Darino for technical assistance.

## Author Contributions

**Conceptualization:** Fernando Navarrete, Armin Djamei.

**Funding acquisition:** Armin Djamei.

**Investigation:** Fernando Navarrete, Nenad Grujic, Alexandra Stirnberg, Indira Saado, David Aleksza, Hazem Adi, André Alcântara, Mamoona Khan.

**Methodology:** Fernando Navarrete, Marco Trujillo, Armin Djamei.

**Project administration:** Fernando Navarrete, Armin Djamei.

**Resources:** Fernando Navarrete, Nenad Grujic, Alexandra Stirnberg, Indira Saado, Michelle Gallei, Hazem Adi, Janos Bindics, Marco Trujillo, Armin Djamei.

**Writing – original draft:** Fernando Navarrete.

**Writing – review & editing:** Fernando Navarrete, Armin Djamei.

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
