## [Decision Letter · Decision Letter 0]

19 Jan 2021

Dear Dr Djamei,

Thank you very much for submitting your manuscript "The Pleiades are a cluster of fungal effectors that inhibit host defenses" for consideration at PLOS Pathogens. As with all papers reviewed by the journal, your manuscript was reviewed by members of the editorial board and by several independent reviewers. All three reviewers, whilst expressing interest in your work, have significant criticisms that require additional experiments to be performed. In light of the reviews (below this email), we would like to invite the resubmission of a significantly-revised version that responds to each of the reviewers' comments.  

We cannot make any decision about publication until we have seen the revised manuscript and your response to the reviewers' comments. Your revised manuscript is also likely to be sent to reviewers for further evaluation.

Sincerely,

Paul Birch

Associate Editor

PLOS Pathogens

Bart Thomma

Section Editor

PLOS Pathogens

Kasturi Haldar

Editor-in-Chief

PLOS Pathogens

orcid.org/0000-0001-5065-158X

Michael Malim

Editor-in-Chief

PLOS Pathogens

orcid.org/0000-0002-7699-2064

Reviewer's Responses to Questions

**Part I - Summary**

Reviewer #1: The authors describe a maize smut fungus gene cluster that encodes 10 secreted effector candidate proteins, the “Pleiades”. They show that most of the Pleiades are able to suppress pattern triggered ROS in both monocots and dicots, and as such the Pleiades are highly interesting to study. Evidence suggest that these secreted proteins suppress ROS by interfering with different steps in this PTI response. As the authors argue, the resulting redundancy between the effector candidates is likely to be essential for the fungal pathogen in order to adapt to the plant’s surveillance of pathogens. The most effective ROS-suppressing Pleiades are found to be Tay1 and Mer1, which are localized to the plant cytosol and cytosol/nucleus, respectively. In addition, the authors show that Tay1 needs to be in the cytosol and Mer1 in the nucleus in order to suppress PTI ROS. By Y2H and IP, it is found that Mer1 interacts with maize, N. benthamiana and Arabidopsis RFI2 E3 ubiquitin ligases, and by an Arabidopsis mutant study, AtRFI2A and AtRFI2B are shown to be required for normal PTI ROS. Finally, in vitro studies suggest that Mer1 triggers auto-ubiquitination and subsequent MG132-sensitive degradation of AtRFI2A. According to the authors, this “indicate” that Mer1 mediates degradation of this E3 ligase in the plant and thereby suppress PTI ROS.

The topic and level of this manuscript is timely and highly relevant for the readers of PLOS Pathogen, as a profound understanding of effector function may well lead to a strategy towards plant disease control. The manuscript is overall very well written and easy to follow, and the experiments are well performed and sound, all reflecting the solid experience of this research team.

Reviewer #2: In this interesting manuscript, the authors identified pleiades as a cluster of fungal effectors that inhibit host defenses. The authors identify eight effectors from this locus that suppress PAMP triggered host ROS accumulation, likely through distinct mechanisms. It is nice to take the advantages of this fungal system to genetically validate the roles of effector/effector cluster.

There are some preliminary data here including the in vivo and in vitro binding between effector and host targets. However, I am concerned that laking detailed and robust mechanistic studies of how effector manipulates plant immunity make the paper is not strong enough to be published in PLOS Pathogens.

Reviewer #3: In this manuscript submitted by Navarrete et al., the authors attempted to elucidate the function of effector proteins encoded on a gene cluster in the genome of the corn smut pathogen Ustilago maydis. The gene cluster named as “Pleiades” encodes 10 putative effector genes and contributes to virulence. However, the function of respective effectors is mostly unknown. The authors mainly focused on two effectors Tay1 and Mer1, of which deletion mutant shows statistically significant reduction of virulence. Additional phenotype of cluster mutant is accumulation of reactive oxygen species (ROS) around fungal hyphae inside infected leaf tissues. Therefore, the authors attempted to link the function of effectors with ROS production in planta. Using transient expression system in N. benthamiana and maize, it is demonstrated that Tay1 and Mer1 can suppress ROS production in planta. Furthermore, plant E3 ubiquitin ligase RFI2 is identified as interaction partner of effector protein through yeast two-hybrid system. Mer1 protein interacts with RFI2 and enhance the auto-ubiquitination activity of RFI2. In Arabidopsis, RFI2 seems to be involved in ROS production.

In my view, the work is carefully performed and the data is clear to demonstrate the function of effectors. While the structure of manuscript is logically organized, I still have a couple of concerns in the current form of manuscript. I describe those concerns below and would like the authors to answer it.

**Part II – Major Issues: Key Experiments Required for Acceptance**

Reviewer #1: I only have one major issue, which concerns the final interpretation of how Mer1 functions, which is primarily based on the in vitro experiments on auto-ubiquitination of the E3 ligase and its degradation. The authors conclude by saying that the data “indicate” and they leave no doubt that Mer1 inhibits RFI2 (l. 333-336). There could certainly be other unfound mechanisms explaining how Mer1 works. Therefore, I should like to see mechanistic studies of this in planta. One could test whether the ROS suppressing function of Mer1 is dependent on AtRFI2A in Arabidopsis. This could easily be addressed by crossing a Mer1 overexpressing line from Fig. 5a with the rfi2A mutant. The resulting Mer1 OE rfi2A line should have a ROS response indistinguishable from the parents. Even though such a result only would show that Mer1 affects the same pathway as RFI2, it would provide further support for the authors interpretation.

Reviewer #2: I have the concerns as the following:

The authors state that pleiades cluster encodes ten proteins, why the authors only choose UMAG_03752 and UMAG_03753 for further investigation. The protein expression of mCherry-Mer1 and SP-mCherry-Mer1 in Fig. 1b should support with WB results. It’s better to use yeast invertase secretion assay to validate the predicted signal peptide of UMAG_03752 and UMAG_03753.

The authors expressed ten proteins in N. benthamiana and tested their ability to suppress PAMP-triggered ROS production, the authors should check whether these proteins are stable by using WB. Is there any way that we could understand the function of Tay1 and Mer1 in planta? e.g. Could they promote infection ?

At least, one of the either BIFC or Pull down data is required to confirm the interactions between effectors and RFI2 homologs.

The authors state that single knockouts (rfi2A and rfi2B) and the double knock out (rfi2A/rfi2B) plants showed a reduced ROS burs. Whether these mutants display clear phenotypes in real infection?

Reviewer #3: In Figure 1, the authors showed the secretion of effector protein by fusion with mCherry protein and microscopy. Secretion of effector protein is also supported by the data in Fig S1, which shows the detection of full-length protein in culture supernatant. However, it is not clear whether the secreted effector is full-length in planta. If the effector proteins lack cysteine residues providing stable protein folding, the effector might be unstable in apoplastic space. Immunoprecipitation and western blot of effector proteins from infected leaf tissues would be helpful to clarify this point.

In Figure 2, the authors showed DAB staining in the leaves infected with U. maydis wild type and cluster mutant to claim ROS accumulation in apoplastic space. Indeed, the hyphae of cluster mutant shows staining along with hyphae, while the wild type does not show it at all. However, the staining pattern looks a little bit weird for me, so I am not sure whether ROS actually accumulated in apoplastic space. To my knowledge, ROS accumulation pattern upon fungal infection is generally seen as accumulation in entire plant cell or as halo at infection sites. When I enlarged a picture, I could see some speckles of DAB staining inside fungal hyphae, which makes me doubt that ROS production actually comes from fungal side (by stresses due to plant defenses, for example). One possible experiment to clarify this point might be a gene silencing of plant genes related to ROS production (e.g. NADPH oxidase) in infected leaf tissues and microscopy to observe whether DAB staining is present or not. Another is to monitor the expression of plant genes related to ROS production upon cluster mutant infection (this would be much easier to imply that ROS production is from plant side).

Last and most major concern is the biological relevance of RFI2 and U. maydis infection in maize. It is very nice result that Mer1 promotes auto-ubiquitination of RFI2 and RFI2 contributes to ROS production in Arabidopsis. However, I miss that the authors do not show whether RFI2 actually contributes to ROS production and is involved in plant immunity upon U. maydis infection in maize. Straightforward experiment to demonstrate could be gene silencing of RFI2 in maize and evaluate fungal virulence to such gene-silenced maize plants, since FoMV-based virus gene silencing should be available in maize.

**Part III – Minor Issues: Editorial and Data Presentation Modifications**

Reviewer #1: Minor issues:

• Please discuss how the E3 degradation may prevent ROS production.

• Care should be taken to distinguish genes and proteins. Examples: In l. 33, “a cluster of ten effectors” should be a “cluster of ten effector genes”. Same in l. 346.

• The Materials and Methods could do with a careful make-over. Many mistakes appear in the text, eg. regarding when to spell numbers or use numerals

Reviewer #2: It’s better to use a well-known nuclear protein as a nuclear marker instead of GFP-NLS. Also, a bright image is required to show the cells in fig. 3b. The cellular localization of Tay1 and Mer1 fusing with Myc/Myr/NES/NLS-mCherry should provide. Also, WB of these proteins should be provided.

The authors state that BD-Tay did not lead to the identification of any reproducible interactors. But why BD-Tay was able to interact with some of the RFI2 homologs in Y2H? Please explain!

Although I am not a native speaker, I think the manuscript need to be carefully polished.

Reviewer #3: (No Response)

PLOS authors have the option to publish the peer review history of their article (what does this mean?). If published, this will include your full peer review and any attached files.

Reviewer #1: No

Reviewer #2: No

Reviewer #3: No
---

## [Decision Letter · Decision Letter 1]

6 May 2021

Dear Dr Djamei,

Thank you very much for submitting your manuscript "The Pleiades are a cluster of fungal effectors that inhibit host defenses" for consideration at PLOS Pathogens. As with all papers reviewed by the journal, your manuscript was reviewed by members of the editorial board and by several independent reviewers. The reviewers appreciated the attention to an important topic. Based on the reviews, we are likely to accept this manuscript for publication, providing that you modify the manuscript according to the review recommendations.

Sincerely,

Paul Birch

Associate Editor

PLOS Pathogens

Bart Thomma

Section Editor

PLOS Pathogens

Kasturi Haldar

Editor-in-Chief

PLOS Pathogens

orcid.org/0000-0001-5065-158X

Michael Malim

Editor-in-Chief

PLOS Pathogens

orcid.org/0000-0002-7699-2064

Reviewer Comments (if any, and for reference):

Reviewer's Responses to Questions

**Part I - Summary**

Reviewer #2: none

Reviewer #3: This is the revised version of the manuscript that was previously submitted by Navarrete et al. The authors identified the virulence effectors of the corn smut pathogen Ustilago maydis suppressing ROS production and also identified plant interaction partner protein. In the revised version, some additional data and text are incorporated.

**Part II – Major Issues: Key Experiments Required for Acceptance**

Reviewer #2: I think the reviewers already addressed my major concerns by either adding additional experiments or reply with strong argument.

Reviewer #3: My first major point was about protein integrity of mCherry-fused effectors in planta. This point is now clarified by additional western blot experiment in infected leaf tissues. For other points, it is pity that the silencing experiments in maize seems to be tricky so that the authors may no be able to obtain useful data. Nevertheless, the results in Arabidopsis somehow supports the function of effectors.

**Part III – Minor Issues: Editorial and Data Presentation Modifications**

Reviewer #2: I am find with their reply on my minor issues.

Reviewer #3: I found some minor issues (see below) in the revised manuscript and would like to ask the authors to correct them.

Page 2, Line 31: “Ustilago maydis” should be italic.

Page 2, Line 34: “in planta” should be italic.

Page 3, Line 50: “pleiades” should be italic (only for p).

Page 7, Line 157: “SPtay1mCherry-Tay1” is better to be SPtay1-mCherry-Tay1, to be consistent with “SPmer1-mCherry-Mer1” from line 156. Same for page 27, Line 792.

Page 10, Line 257: For the description of high stringency media, I suggest not to use abbreviated letters like LWHA when it appears at first place in text.

Page 12, Line 300: Through the entire manuscript, the word “knockout” or “knock out” should be consistent (e.g. Line 304, “knockouts” and “knock out”).

Page 13, Line 330: “our results indicate, that” I don’t think a comma goes there.

Page 19, Line 501: “Einheitserde” I assume that this is commercial name. Could you provide information?

Page 28, Line 855: “U. maydis” should be italic.

Page 29, Line 861, 862: “SPMer1mCherry-Mer1” is better to be SPMer1-mCherry-Mer1.

Page 29, Line 875: “H202” should be “H2O2”.

Page 29, Line 899: There is period two times after “50µm”.

Figure 6: For broad readers, the schematic picture is hard to understand what it actually depicts. Please reconsider it and provide more professional figure and description in legend.

PLOS authors have the option to publish the peer review history of their article (what does this mean?). If published, this will include your full peer review and any attached files.

Reviewer #2: No

Reviewer #3: No

Figure Files:

Data Requirements:

Reproducibility:

References:

---

## [Editor Report · Decision Letter 2]

13 May 2021

Dear Dr Djamei,

We are pleased to inform you that your manuscript 'The Pleiades are a cluster of fungal effectors that inhibit host defenses' has been provisionally accepted for publication in PLOS Pathogens.

Best regards,

Paul Birch

Associate Editor

PLOS Pathogens

Bart Thomma

Section Editor

PLOS Pathogens

Kasturi Haldar

Editor-in-Chief

PLOS Pathogens

orcid.org/0000-0001-5065-158X

Michael Malim

Editor-in-Chief

PLOS Pathogens

orcid.org/0000-0002-7699-2064
---

## [Editor Report · Acceptance letter]

3 Jun 2021

Dear Dr Djamei,

We are delighted to inform you that your manuscript, "The Pleiades are a cluster of fungal effectors that inhibit host defenses," has been formally accepted for publication in PLOS Pathogens.

Best regards,

Kasturi Haldar

Editor-in-Chief

PLOS Pathogens

orcid.org/0000-0001-5065-158X

Michael Malim

Editor-in-Chief

PLOS Pathogens

orcid.org/0000-0002-7699-2064